# Cryo-EM structure of the volume-regulated anion channel LRRC8D isoform identifies features important for substrate permeation

Ryoki Nakamura[1,11], Tomohiro Numata [2,11], Go Kasuya [3,11✉], Takeshi Yokoyama[4], Tomohiro Nishizawa [1], Tsukasa Kusakizako [1], Takafumi Kato[1], Tatsuya Hagino[1], Naoshi Dohmae[5], Masato Inoue[6], Kengo Watanabe [6], Hidenori Ichijo [6], Masahide Kikkawa [7], Mikako Shirouzu [4], Thomas J. Jentsch [8], Ryuichiro Ishitani[1], Yasunobu Okada[9,10✉] & Osamu Nureki [1✉]

Members of the leucine-rich repeat-containing 8 (LRRC8) protein family, composed of the five LRRC8A-E isoforms, are pore-forming components of the volume-regulated anion channel (VRAC). LRRC8A and at least one of the other LRRC8 isoforms assemble into heteromers to generate VRAC transport activities. Despite the availability of the LRRC8A structures, the structural basis of how LRRC8 isoforms other than LRRC8A contribute to the functional diversity of VRAC has remained elusive. Here, we present the structure of the human LRRC8D isoform, which enables the permeation of organic substrates through VRAC. The LRRC8D homo-hexamer structure displays a two-fold symmetric arrangement, and together with a structure-based electrophysiological analysis, revealed two key features. The pore constriction on the extracellular side is wider than that in the LRRC8A structures, which may explain the increased permeability of organic substrates. Furthermore, an N-terminal helix protrudes into the pore from the intracellular side and may be critical for gating.

[1] Department of Biological Sciences, Graduate School of Science, The University of Tokyo, 7-3-1 Hongo, Bunkyo-ku, Tokyo, Japan. [2] Department of Physiology, Graduate School of Medical Sciences, Fukuoka University, 7-45-1 Nanakuma, Johnan-ku, Fukuoka, Japan. [3] Division of Integrative Physiology, Department of Physiology, Jichi Medical University, 3311-1 Yakushiji, Shimotsuke-shi, Tochigi, Japan. [4] Laboratory for Protein Functional and Structural Biology, RIKEN Center for Biosystems Dynamics Research, 1-7-22 Suehiro-cho, Tsurumi-ku, Yokohama-shi, Kanagawa, Japan. [5] Biomolecular Characterization Unit, RIKEN Center for Sustainable Resource Science, 2-1 Hirosawa, Wako-shi, Saitama, Japan. [6] Laboratory of Cell Signaling, Graduate School of Pharmaceutical Sciences, The University of Tokyo, 7-3-1 Hongo, Bunkyo-ku, Tokyo, Japan. [7] Department of Cell Biology and Anatomy, Graduate School of Medicine, The University of Tokyo, 7-3-1 Hongo, Bunkyo-ku, Tokyo, Japan. [8] Physiology and Pathology of Ion Transport, Leibniz-Forschungsinstitut für Molekulare Pharmakologie (FMP) and Max-Delbrück-Centrum für Molekulare Medizin, D-13125 Berlin, Germany. [9] Department of Molecular Cell Physiology, Kyoto Prefectural University of Medicine, Kyoto, Japan. [10] Department of Cell Physiology, National Institute for Physiological Sciences, Okazaki, Japan. [11]These authors contributed equally: Ryoki Nakamura, Tomohiro Numata, Go Kasuya. ✉email: gokasuya@jichi.ac.jp; okada@nips.ac.jp; nureki@bs.s.u-tokyo.ac.jp

In vertebrates, cell volume is regulated by various membrane transporters and channels, which protect against intracellular and extracellular osmotic alterations[1]. The volume-regulated anion channel (VRAC), also called the volume-sensitive outwardly rectifying anion channel or volume-sensitive organic osmolyte-anion channel, is an anion channel activated upon cell swelling. The VRAC-mediated transport of chloride ions and small organic compounds, including osmolytes such as taurine or glutamate, drives water efflux to counteract cell swelling[2,3]. The leucine-rich repeat-containing 8 (LRRC8) protein family, composed of the five isoforms LRRC8A to LRRC8E, was recently identified as a pore-forming component of the VRAC[4,5]. Biochemical and biophysical analyses demonstrated that heterohexamers containing at least one LRRC8A isoform and at least one of the other four LRRC8 isoforms (LRRC8B-LRRC8E) form functional VRAC channels[4,6–8]. The isoform combinations forming LRRC8 hetero-hexamers underlie the functional diversity of the channel properties, including open probability[7], gating kinetics[4,9], and substrate specificity[8,10]. However, LRRC8A homo-hexamers can still form functional, although not physiological, channels with much smaller swelling-induced currents, when overexpressed in HEK293 cells with the disruption of all five human *LRRC8* genes[11,12]. Among the five LRRC8 isoforms, the LRRC8D isoform is the largest one owing to a long first extracellular loop between the transmembrane (TM) 1 and 2 helices[13]. When incorporated into LRRC8 hetero-hexamers, the LRRC8D isoform facilitates the permeation of uncharged substrates such as taurine and GABA, negatively charged glutamate, and even positively charged lysine[8,10]. Moreover, LRRC8D-containing hetero-hexamers facilitate the uptake of the antibiotic blasticidin S[14] and the Pt-based anticancer drug cisplatin[10], resulting in cancer drug resistance in patients with down-regulated LRRC8D isoforms[10].

The recently determined single-particle cryo-electron microscopy (cryo-EM) structures of the human and mouse LRRC8A homo-hexamers in either detergent or lipid nanodiscs, as well as the lower resolution structure of mouse LRRC8A/C hetero-hexamers in detergent, provided the structural framework for hexamer formation, ion permeation, and recognition of a VRAC-specific inhibitor in the LRRC8 protein family[11,15–17]. The ion-selective pore of LRRC8 hexamers is located along the central axis and is formed by the N-terminal half of each of the six subunits, with a similar topology to the pannexin and gap junction channels, including connexins and innexins[18–22]. Recent electrophysiological analyses demonstrated that the first several N-terminal residues preceding the TM1 helix[16,23] and the first extracellular loop (EL1) between the TM1 and TM2 helices[9,12] are involved in the diversity of channel properties, including ion permeability and selectivity as well as the kinetics of inactivation. In all previously determined LRRC8 structures, however, the structures of both the N-terminal and EL1 regions remained unresolved due to their flexibility[11,15–17]. In addition, the structural basis of how LRRC8 isoforms other than LRRC8A contribute to the ion permeation and diversity of VRAC functions has remained elusive, due to the lack of high-resolution structures of other LRRC8 isoforms. The LRRC8D isoform is particularly interesting, because its incorporation into heteromers is requisite for the permeation of many organic substrates, independent of their electrical charge. In this study, we determined the structure of the human LRRC8D homo-hexamer, including the N-terminal region, by cryo-EM single-particle analysis, and conducted a structure-based electrophysiological analysis to investigate the mechanism underlying the contribution of the LRRC8D isoform to the LRRC8 protein function.

## Results

**Structure determination of human LRRC8D protein.** When expressed in Sf9 insect cells from *Spodoptera frugiperda*, which lacks endogenous *LRRC8* genes[13], fluorescence-detection size-exclusion chromatography[24] of the C-terminally EGFP-tagged human LRRC8D (HsLRRC8D; the "Hs" refers to *Homo sapiens*) protein showed a sharp and monodisperse peak with a similar elution volume to that of the human LRRC8A (HsLRRC8A) protein, with structures determined by cryo-EM single-particle analysis[15,16], indicating that the HsLRRC8D protein is a promising candidate for structural analysis (Supplementary Fig. 1a). A mass spectrometry analysis of the purified HsLRRC8D protein sample from Sf9 insect cells confirmed that the sample contains no LRRC8 isoforms other than HsLRRC8D and that the HsLRRC8D protein exists as a homomer (Supplementary Fig. 1b, c; Supplementary Data 1). To assess the functional properties of HsLRRC8D, we performed whole-cell patch-clamp analyses using HEK293 cells, in which all five human *LRRC8* genes (*LRRC8[−/−]* cells) were disrupted and swelling-induced currents were not observed (Supplementary Fig. 1d, e)[4]. As shown in Supplementary Fig. 1d, e and consistent with previous reports[4,11,12], *LRRC8[−/−]* cells co-transfected with the HsLRRC8A wild type and HsLRRC8D wild type plasmids (HsLRRC8A/D) showed swelling-induced currents, and those transfected with the HsLRRC8A wild type plasmid (HsLRRC8A) alone showed slight swelling-induced currents. *LRRC8[−/−]* cells transfected with only the HsLRRC8D wild type plasmid (HsLRRC8D) showed no swelling-induced currents, confirming the notion that the functional VRAC channels are heteromers including the LRRC8A isoform as a crucial component[4]. Since the uncertain and most likely variable stoichiometry of the heteromeric LRRC8A/D channels greatly complicates the determination of their structures, we opted to elucidate the structure of HsLRRC8D homomers for comparison with the previously determined structures of LRRC8A homo-hexamers. Even if these homomers are non-functional in cells under physiological conditions, they may reveal the structural differences between LRRC8 isoforms that underlie the functional diversity of VRACs. Therefore, we attempted to elucidate the structure of the HsLRRC8D homomer.

The purified HsLRRC8D protein was vitrified on grids, and images were recorded by electron microscopy (Supplementary Fig. 2b). The final 3D map was reconstructed to an overall 4.36 Å resolution (local resolution of the transmembrane region is ~3.9 Å) with C2 symmetry imposed, and contained six subunits of HsLRRC8D (hereafter referred to as α to ζ subunits) (Supplementary Fig. 2a, c–f; Supplementary Table 1). Using the previously determined structures of LRRC8A homo-hexamers in detergent (PDB 5ZSU and 6G9O) as a guide, we assigned the topologies of the HsLRRC8D regions, including the extracellular, transmembrane (TM), intracellular, and leucine-rich repeat (LRR) regions (Supplementary Fig. 2e). Data collection, refinement, and validation statistics for the cryo-EM structure are shown in Table 1.

**Overall structure.** Overall, HsLRRC8D adopts a homo-hexameric architecture composed of four regions: extracellular, TM, and intracellular regions in the N-terminal half of each subunit, and a leucine-rich repeat (LRR) region contained in the C-terminal half (Fig. 1). The overall HsLRRC8D structure is largely consistent with the three previously determined LRRC8A structures in detergent[11,15,16] or nanodiscs[17], with some exceptions (Fig. 1; Supplementary Figs. 3 and 4a–e). Among the previously reported LRRC8A structures, unless otherwise noted, we mainly used our human LRRC8A structure (PDB ID: 5ZSU)[15] for

**Table 1 Cryo-EM data collection, refinement, and validation statistics.**

|  | HsLRRC8D (EMD-30029) (PDB 6M04) |
| --- | --- |
| *Data collection and processing* |  |
| Magnification[a] | x23,500 |
| Voltage (kV) | 200 |
| Electron exposure (e–/Å$^2$) | 50 |
| Defocus range (μm) | 0.5–2.5 |
| Pixel size (Å) | 1.490 |
| Symmetry imposed | C2 |
| Initial particle images (no.) | 2,056,807 |
| Final particle images (no.) | 247,154 |
| Map resolution (Å) | 0.143 |
| FSC threshold |  |
| Map resolution range (Å) | 3.9–6.0 |
| *Refinement* |  |
| Initial model used (PDB code) | 5ZSU, 6G9O |
| Model resolution (Å) | 0.5 |
| FSC threshold |  |
| Model resolution range (Å) | 4.36 |
| Map sharpening *B* factor (Å$^2$) | −211.2 |
| Model composition |  |
| Non-hydrogen atoms | 34,666 |
| Protein residues | 4282 |
| *B* factors (Å$^2$) |  |
| Protein | 106.9 |
| R.m.s. deviations |  |
| Bond lengths (Å) | 0.007 |
| Bond angles (°) | 1.288 |
| Validation |  |
| MolProbity score | 2.38 |
| Clashscore | 10.11 |
| Poor rotamers (%) | 2.06 |
| Ramachandran plot |  |
| Favored (%) | 87.11 |
| Allowed (%) | 12.85 |
| Disallowed (%) | 0.05 |

[a]Calibrated pixel size at the detector.

comparison with HsLRRC8D, since the structure determination conditions are the most similar. The overall HsLRRC8D structure displays a twofold symmetric "dimer of trimers" arrangement; in other words, the structures of the α and δ subunits, those of the β and ε subunits, and those of the γ and ζ subunits are identical (Figs. 1 and 2). In contrast, the overall assembly of LRRC8A homomers in detergent displayed a threefold symmetric "trimer of dimers" arrangement, whereas LRRC8A homomers in nano-discs displayed a sixfold symmetric arrangement in the TM region and conformational heterogeneity in the LRR region, due to its flexibility (Figs. 1 and 2; Supplementary Fig. 3)[11,15–17]. The previous comparisons of LRRC8A structures in either detergent or nanodiscs suggested that detergent micelles play an important role in promoting the threefold symmetric arrangement of the TM region, whereas nanodiscs with an included lipid bilayer promote the sixfold symmetric arrangement[17]. Our HsLRRC8D structure revealed that, in addition to the threefold and sixfold symmetries, LRRC8 hexamers can form twofold symmetry. However, further studies are required to confirm whether these structural arrangements occur in the lipid-embedded heteromers containing the LRRC8A isoform; i.e., under physiological conditions.

Each subunit of HsLRRC8D has two extracellular loops (EL1 and EL2), four transmembrane helices (TM1-4), and two intracellular loops (IL1 and IL2) in the N-terminal half, and an LRR domain that contains the leucine-rich repeat N-terminal helix (LRRNT), 15 leucine-rich repeats (LRR1–15), and the following leucine-rich repeat C-terminal helix (LRRCT) in the C-terminal half (Fig. 1b). Notably, we discovered an additional N-terminal helix (NTH) (Fig. 1b; Supplementary Figs. 4a and 6), which was not observed in the previously reported LRRC8A structures (Supplementary Fig. 4b–e)[11,15–17]. It is formed by the N-terminal residues preceding the TM1 helix and protrudes into the channel pore, as in the previous structures of the connexin[18,19,25] and innexin[20,22] channels (Supplementary Fig. 5a, b), which show weak homology to LRRC8 proteins (Fig. 1b; Supplementary Figs. 4a–e and 6)[13].

**Pore constriction**. The channel pore of the HsLRRC8D structure is located along the central axis perpendicular to the cell

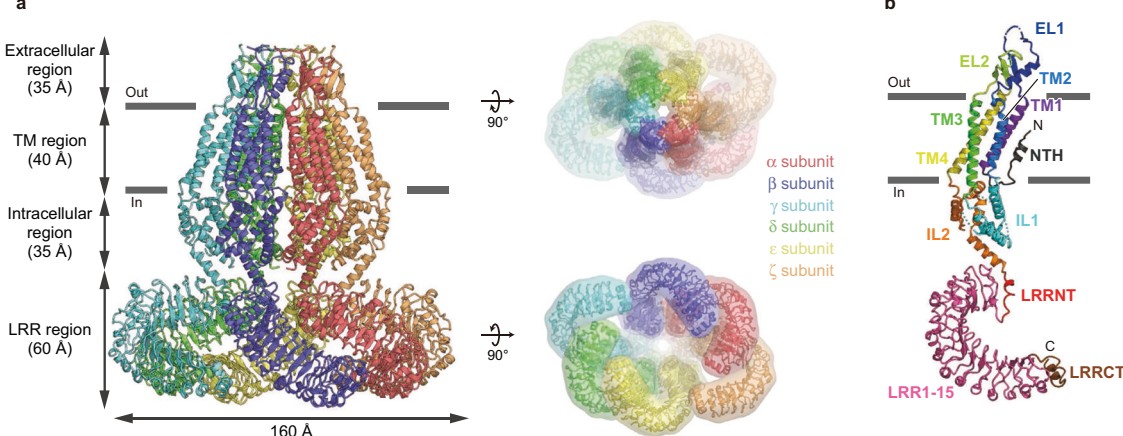

**Fig. 1 Overall structure and subunit interactions of the HsLRRC8D hexamer. a** The overall structure of the HsLRRC8D hexamer, viewed parallel to the membrane (left) and from the extracellular (right, upper) and intracellular (right, lower) sides. **b** A subunit of the HsLRRC8D structure, viewed parallel to the membrane. According to the previously proposed HsLRRC8A hexamer structure (PDB ID: 5ZSU)[15], each region is colored as follows: NTH dark gray, TM1 purple, EL1 blue, TM2 light blue, IL1 cyan, TM3 green, EL2 light green, TM4 yellow, IL2 orange, LRRNT red, LRR1-15 pink, and LRRCT brown. The N- and C-termini are indicated by 'N' and 'C', respectively. The molecular graphics were illustrated with CueMol (http://www.cuemol.org/).

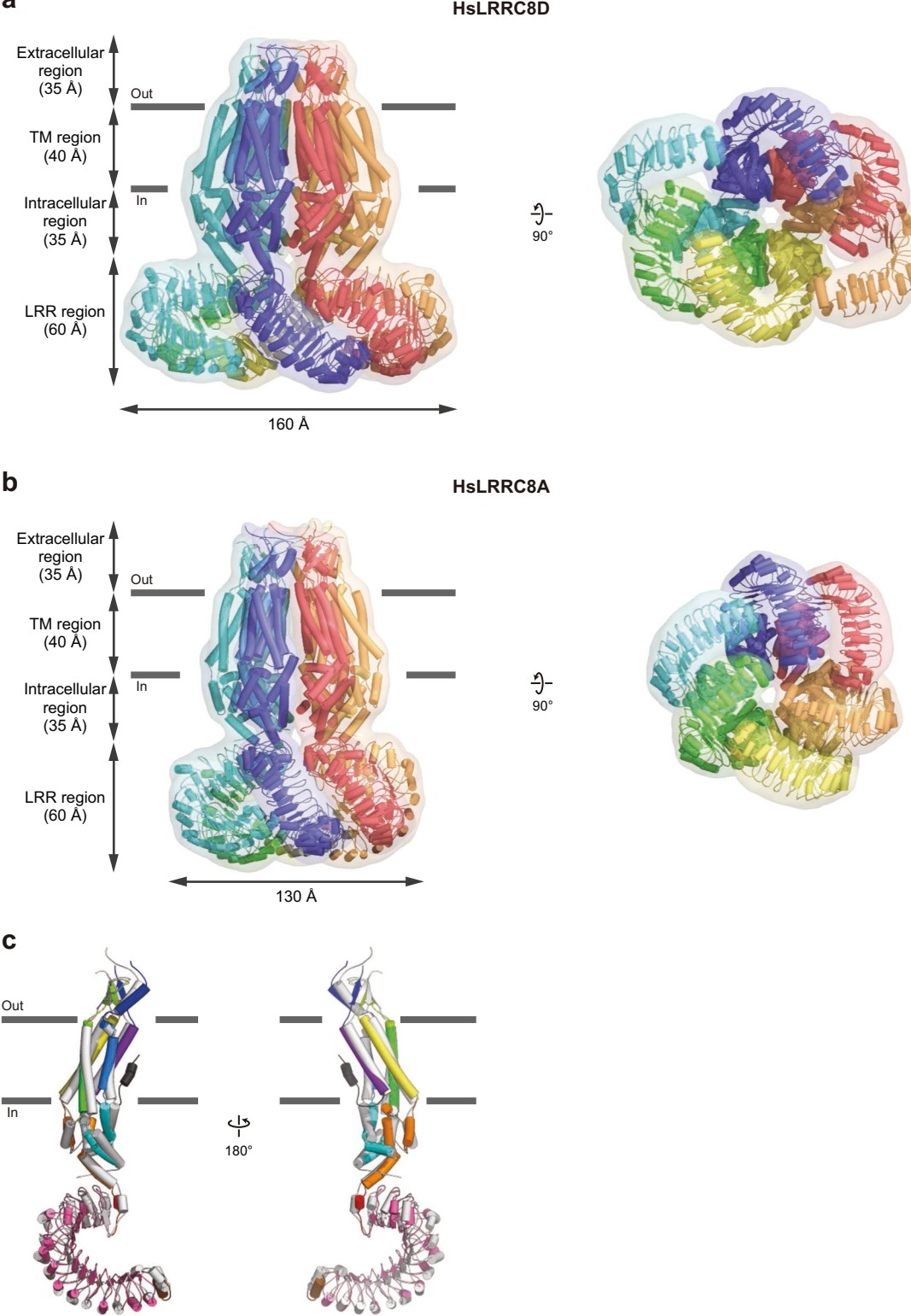

**Fig. 2 Comparisons of the HsLRRC8D and HsLRRC8A structures. a, b** Overall structures of the HsLRRC8D (**a**) and HsLRRA8A (PDB ID: 5ZSU)[15] (**b**) hexamers, viewed parallel to the membrane (left) and from the intracellular side (right). **c** Superimposition of the HsLRRC8D (colored according to Fig. 1b) and the HsLRRC8A (gray) structures, using the Cα atoms of the protomers. The structures are depicted in cylinder representations with molecular surfaces.

membrane. In agreement with the previously determined LRRC8A structures, it is constricted at the N-terminal tip of a helix (EL1H) on the extracellular side (Fig. 3a–c)[11,15–17]. The sequence alignment predicted that the most constricted site is formed by the Phe143 residue, as its aromatic ring restricts the pore diameter to 11.5 Å (Fig. 3b–d; Supplementary Fig. 4a). Notably, this value is close to the previously reported native VRAC pore diameters of 11.4-14.2 Å, estimated by measuring calixarene conductance in endothelial cells[26], and 12.6 Å obtained by differential polymer partitioning in epithelial cells[27]. In contrast, the pore diameters are 7.6–9.6 Å in the previously determined LRRC8A structures[11,15–17], in which the corresponding residue is replaced by Arg103, with its positively charged side chain. Hence, the pore diameter at the most constricted external site is narrower in LRRC8A than in LRRC8D homomers (Fig. 3b, c; Supplementary Fig. 4a–e). Considering that LRRC8 proteins physiologically form hetero-hexamers, this difference in the pore diameters at the most constricted site between the LRRC8D and LRRC8A homomer structures suggests that the inclusion of the LRRC8D isoform into LRRC8A-containing heteromers may be structurally involved in the increased permeability of VRACs to organic substrates.

To investigate the functional importance of the residue at the constricted site, we created the HsLRRC8D F143R mutant to yield a heteromeric channel with only arginine at the constricted site, and performed an electrophysiological analysis to assess the permeability of various anions using LRRC8[−/−] cells co-transfected with HsLRRC8A wild type and HsLRRC8D wild type or HsLRRC8D F143R mutant plasmids (HsLRRC8A/D and HsLRRC8A/D (F143R), respectively). As compared to HsLRRC8A/D, the HsLRRC8A/D (F143R) channel showed decreased permeability to negatively charged glutamate and gluconate (Fig. 3e, f). Furthermore, while the HsLRRC8A/D channel showed higher permeability for glutamate than for gluconate, the HsLRRC8A/D (F143R) channel showed similar permeability to glutamate and gluconate (Fig. 3e, f). These results suggested that the mutation from the non-charged phenylalanine to the positively charged arginine changes the electrostatic effects and tightens the pore size at the constricted site on the extracellular side, thus exhibiting a greater effect on tightening the pore size than on changing the electrostatic effect, thereby leading to a reduction, rather than an increase, in the permeability to glutamate and gluconate (Fig. 3f). However, further structural analyses are required to understand the precise effect of the mutation from phenylalanine to arginine, since the structure of the HsLRRC8D F143R mutant has not been determined. Notably, previous reports indicated that the positive charge and the bulkiness of Arg103 may be involved in the anion selectivity filter[11] and the pore blockage by extracellular ATP[16], respectively. Our results suggest that the replacement of Arg103 by Phe143 in LRRC8D may increase the permeability of VRAC to organic compounds, predominantly by an increase in the pore diameter.

**N-terminal helix**. Importantly, unlike the previously determined LRRC8A structures[11,15–17] (Supplementary Fig. 4b–e), we observed an additional N-terminal helix (NTH) located in the TM region innermost to the channel pore (Figs. 3b, d and 4a; Supplementary Figs. 4a and 6), entering the channel pore from the cytoplasmic side, similar to the connexin[18,19,25] and innexin[20,22] structures (Supplementary Fig. 5) and consistent with a previous mutagenesis study of LRRC8/VRAC channels[23]. The NTH is formed by the Ala5 to Asn11 residues at the N-terminal region preceding the TM1 helix and closely interacts with the TM1 helix (Figs. 3b, d and 4a; Supplementary Fig. 4a). In the HsLRRC8D structure, the Leu4, Val7, and Leu10 residues face the channel

pore, whereas the Ala5, Ser9, and Asp12 residues face the TM1 helix (Figs. 3b, d and 4a; Supplementary Figs. 4a and 6). To investigate whether the NTH observed in the HsLRRC8D structure is formed in functional LRRC8 hetero-hexamers, we created a series of cysteine mutants in the N-terminal region of the HsLRRC8D protein, and performed an electrophysiological analysis to monitor the accessibility of the thiol-reactive reagent 2-sulfonatoethyl methane-thiosulfonate (MTSES), using LRRC8[−/−] cells co-transfected with HsLRRC8A wild type and one of the HsLRRC8D mutants. All mutants showed swelling-induced currents (Fig. 4b; Supplementary Data 2). For the mutations at Val7 and Leu10, MTSES reduced the swelling-induced currents, whereas MTSES enhanced them for the mutation of Leu4 (Fig. 4c, d; Supplementary Data 2), demonstrating that these residues are accessible from the aqueous phase. These electrophysiological results are consistent with the HsLRRC8D structure, in which these three residues face the channel pore (Fig. 4a). On the basis of the present data, including the HsLRRC8D structure and the electrophysiological experiments in HsLRRC8A/D channels, as well as the previous electrophysiological results observed in LRRC8A/C, LRRC8A/D, and LRRC8A/E channels[23], we conclude that the N-terminal region in the HsLRRC8D protein serves as a pore-forming helix in LRRC8D homo-hexamers and functional LRRC8A/D hetero-hexamers, similar to the cases of connexin and innexin.

## Discussion
In this study, we present the 4.36 Å resolution cryo-EM structure of the human LRRC8D homo-hexamer, the largest isoform in the LRRC8 protein family, which is important to increase the permeability to organic substrates in VRAC (Fig. 1). The overall assembly of HsLRRC8D adopts the homo-hexameric architecture, consistent with the previously determined LRRC8A homo-hexameric structures[11,15–17]. However, while the HsLRRC8D structure displayed a twofold symmetric "dimer of trimers" arrangement, the LRRC8A structures displayed threefold symmetric "trimer of dimers" or sixfold symmetric arrangements, suggesting the conformational flexibility of the LRRC8 protein family (Fig. 2). Since LRRC8 proteins form hetero-hexamers under physiological conditions, further studies are needed to investigate the subunit–subunit interactions of the functional LRRC8 hetero-hexamers in cells.

As predicted by the sequence alignment[13] and the previously determined LRRC8A structures[11,15–17], the channel pore of the HsLRRC8D structure is located along the central axis and constricted at the N-terminal tip of EL1H on the extracellular side by the Phe143 residue (Fig. 3a–d), which corresponds to the Arg103 residue in the LRRC8A structures (Fig. 3d)[11,15–17]. Notably, the diameter of the most constricted site on the extracellular side is wider in the HsLRRC8D structure than in the LRRC8A structures. Our electrophysiological analysis using the LRRC8D F143R mutant suggested that the uncharged phenylalanine residue plays an important role in the substrate permeability, by adjusting the pore size in the LRRC8 protein family (Fig. 3e, f). The wider external opening of LRRC8D than that of the LRRC8A homomers might underlie the increased permeability of LRRC8D-containing VRACs for organic substrates including negatively charged glutamate and gluconate (Fig. 3f), as well as uncharged substrates such as taurine, GABA, and cisplatin[8,10], although it remains undetermined whether the structure actually represents an open conformation of this channel. In the TM region innermost to the channel pore, an additional NTH preceding the TM1 helix is observed (Figs. 3b–d and 4a; Supplementary Fig. 4a), similar to the connexin and innexin structures in which the NTH is thought to be crucial for channel gating

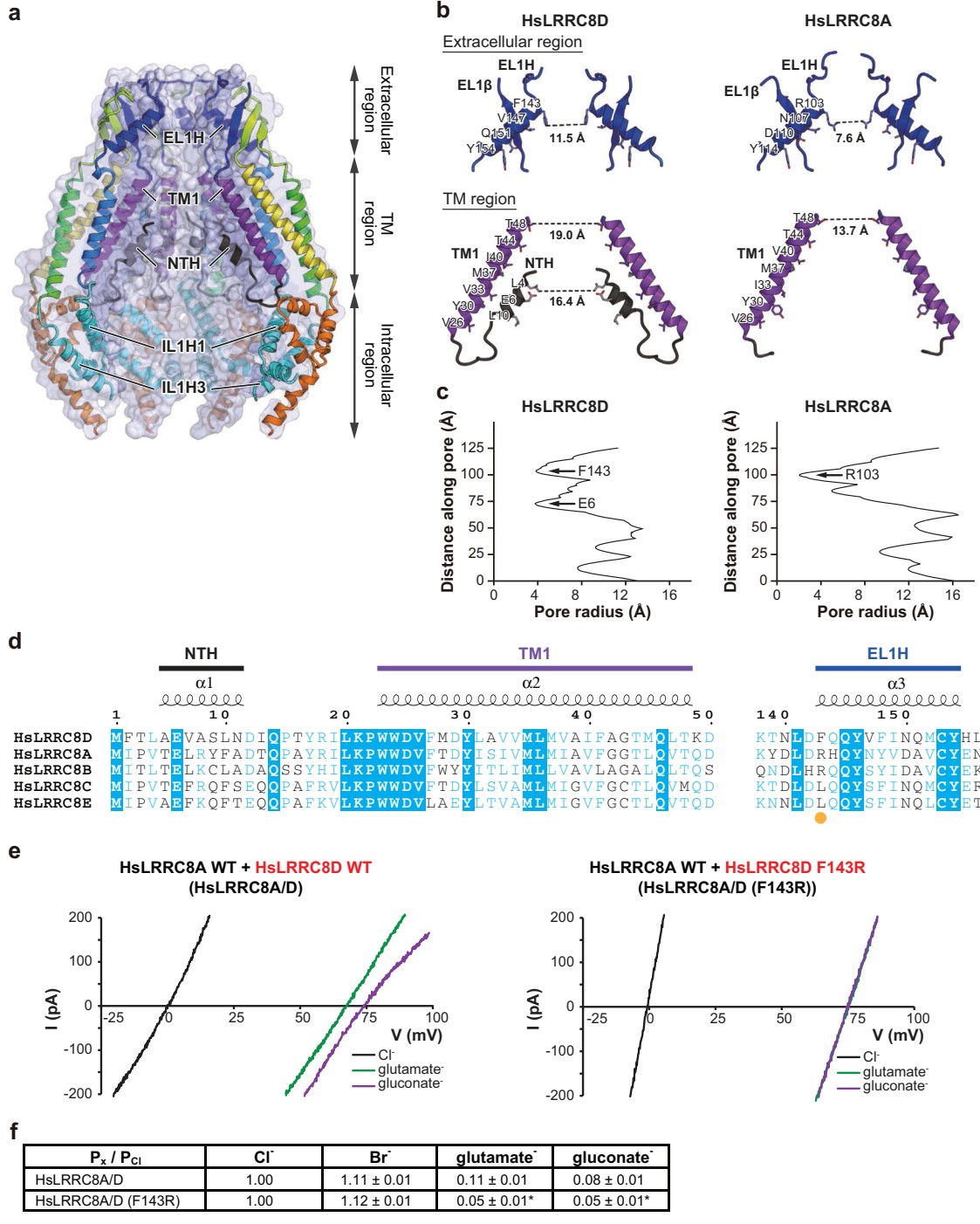

**Fig. 3 Role of pore constriction in anion selectivity. a** Cross-sections of the surface representation for the N-terminal half of the HsLRRC8D subunits, showing the ion channel pore. The four subunits are colored according to Fig. 1b, and are shown in cartoon representations. **b** Close-up views of the pore constrictions formed by the extracellular and TM regions of the HsLRRC8D (left) and HsLRRC8A (PDB ID: 5ZSU)[15] (right) structures. For clarity, only two diagonal subunits viewed parallel to the membrane are shown. The side chains of the pore-lining amino acid residues from NTH, TM1, and EL1H are depicted in stick representations. The distances between the residues involved in the pore constrictions are shown. **c** The pore radii for the HsLRRC8D and HsLRRC8A (PDB ID: 5ZSU)[15] structures. The pore size was calculated with the program HOLE[37]. **d** Sequence alignment of the human LRRC8 isoforms around the pore constriction formed by the extracellular and TM regions. The amino acid sequences were aligned using Clustal Omega[38], and are shown using ESPript3[39]. The NTH, TM1, and EL1H regions of HsLRRC8D are labeled above the alignments. For the sequence alignment, the following human LRRC8 isoforms were used: HsLRRC8A, NCBI Reference sequence number: NP_062540.2; HsLRRC8B, NP_056165.1; HsLRRC8C, NP_115646.2; HsLRRC8D, NP_060573.2; and HsLRRC8E, AAH70089.1. The amino acid residue involved in the pore constriction on the extracellular side is highlighted with an orange dot. **e** Representative current traces of swelling-induced currents in response to voltage ramps of −100 to +100 mV, in one *LRRC8*−/− cell co-transfected with HsLRRC8A wild type and HsLRRC8D wild type (left), and another *LRRC8*−/− cell co-transfected with HsLRRC8A wild type and HsLRRC8D F143R mutant (right). **f** Relative anion selectivity ($P_x/P_{Cl}$) calculated from the shifts of the reversal potential induced by the replacement of chloride ions with other anions. Data are means ± SEM ($n = 5–6$). *$P < 0.05$ compared to the data from *LRRC8*−/− cells co-transfected with HsLRRC8A wild type and HsLRRC8D wild type. *P* values were calculated by the Student's *t* test.

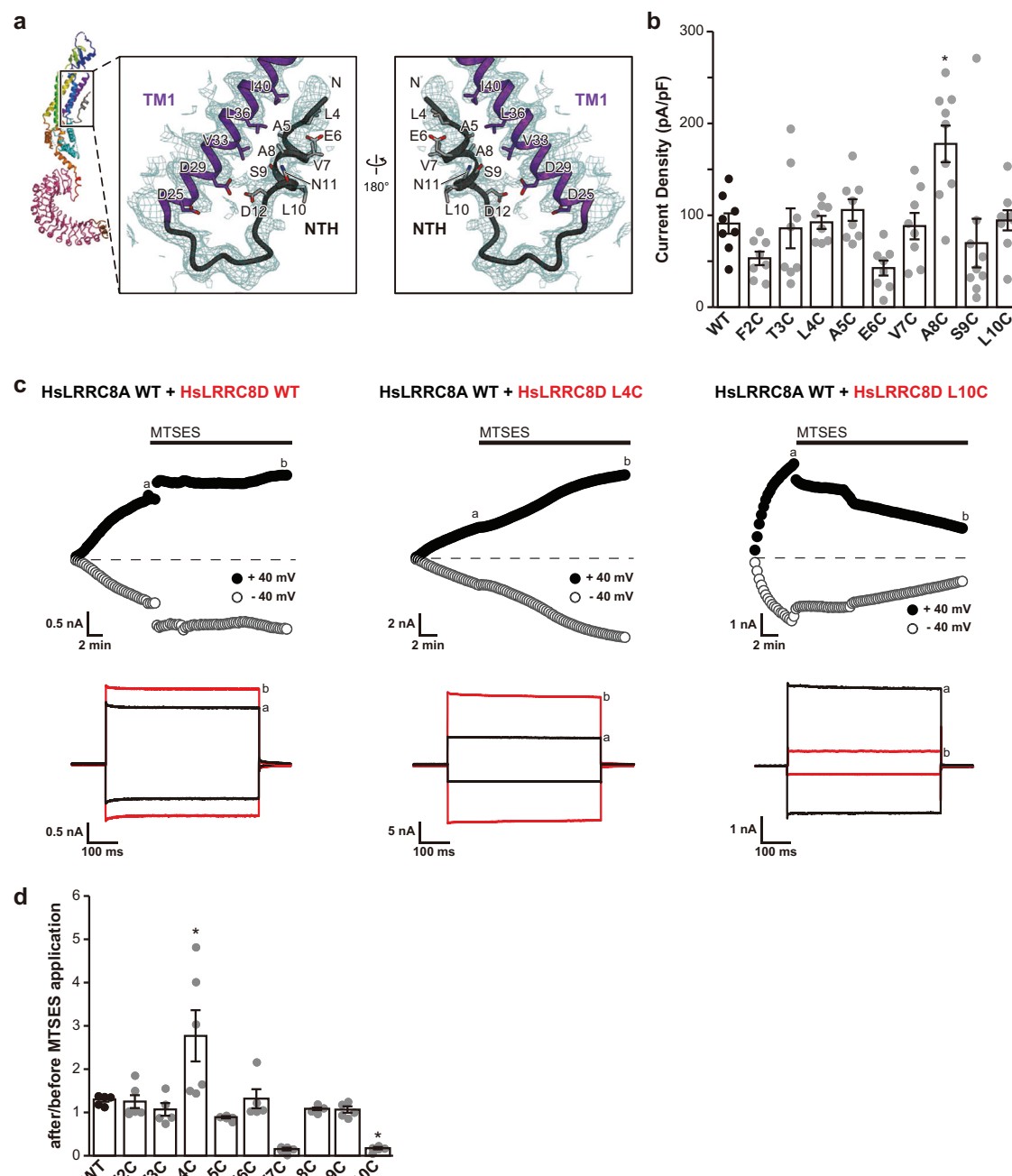

**Fig. 4 Role of the NTH in substrate permeation. a** Close-up view of the NTH and TM1 helix of HsLRRC8D. The EM density map around the NTH and TM1 helix is shown in dark cyan. The side chains of the NTH and TM1 helix are depicted in stick representations. **b** Comparisons of the cell swelling-induced whole-cell currents recorded at +40 mV in $LRRC8^{-/-}$ cells co-transfected with HsLRRC8A wild type and HsLRRC8D wild type, or with HsLRRC8A wild type and each of nine HsLRRC8D cysteine-substituted mutants ($n = 8$–9). **c** Representative time courses recorded from the start of the whole-cell configuration with hypertonic pipette solution (upper panels) and expanded traces of current responses recorded at the time points a and b (lower panels) in the $LRRC8^{-/-}$ cells co-transfected with HsLRRC8A wild type and HsLRRC8D wild type, HsLRRC8A wild type and HsLRRC8D L4C mutant, or HsLRRC8A wild type and HsLRRC8D L10C, during applications (every 15 s) of alternating step pulses of ±40 mV. Black bars indicate an application of 1 mmol L$^{-1}$ MTSES. **d** Swelling-induced currents at +40 mV in the cells expressing HsLRRC8A wild type and HsLRRC8D wild type, or HsLRRC8A wild type and each of nine HsLRRC8D cysteine-substituted mutants after MTSES application, normalized to those before MTSES application. Each column represents the mean ± SEM ($n = 5$–6). *$P < 0.05$ compared to wild type. $P$ values are the results of one-way ANOVA analysis of variance, Tukey–Kramer test.

(Supplementary Fig. 5)[18–21,25]. Further structure-based electrophysiological analyses using a cysteine modifier supported that the NTH observed in the HsLRRC8D structure also exists in the LRRC8A/D hetero-hexamer, with channel entry from the cytoplasmic side. Whether the other LRRC8 proteins also form the NTH observed here remains to be shown. However, a secondary structure prediction of the N-terminal regions of the five LRRC8 isoforms suggested that all LRRC8 isoforms may form the NTH (Supplementary Fig. 4f), and the notion that the LRRC8 N-termini line the VRAC pore is strongly supported by the recent mutagenesis study by Zhou et al.[23]. Using a series of double cysteine mutants of the N-terminal residues in the LRRC8A/C,

LRRC8A/D, and LRRC8A/E channels, Zhou et al.[23] demonstrated that these residues are involved in channel properties, such as ion permeability and kinetics of inactivation. In particular, Zhou et al.[23] found that the conserved Glu6 is important for anion selectivity and thus predicted that the N-terminal regions form part of the channel pore. These results are consistent with the HsLRRC8D structure in which Glu6 also faces the channel pore, as shown in Fig. 4a and Supplementary Fig. 6. Furthermore, Zhou et al.[23] reported that the anion selectivity changes induced by the double cysteine mutations at Glu6 are greater in the LRRC8A/C channels than those in the LRRC8A/D and LRRC8A/E channels. These results suggest that the sequence diversity at the N-terminal region is important for substrate permeability, which might affect the reactivity of MTSES to the mutations at Glu6. Together with the electrophysiological data, the present cryo-EM structure, which is the first to resolve the N-terminus of LRRC8D to our knowledge, now clearly demonstrates that the NTH of LRRC8D participates in forming the channel pore.

In summary, the pore constriction on the extracellular side in LRRC8D homo-hexamers is wider than that in LRRC8A homo-hexamers, and the NTH participates in the formation of the innermost pore by protruding into the intracellular entrance. These results present the structural framework for understanding the mechanism underlying the contribution of the LRRC8D isoform to the functions and properties of VRAC.

## Methods

**Protein expression, purification, and sample preparation for cryo-EM**. Protein expression, purification, and sample preparation were performed as described previously[15], except that Sf9 insect cells from *Spodoptera frugiperda* (ATCC, CRL-1711) were used as the protein expression host. The human LRRC8D wild-type protein (NCBI Reference sequence number: NP_060573.2) was cloned from human brain cDNA (ZYAGEN) into the pFastBac1 vector (Thermo Fisher Scientific), with a C-terminal GFP-His8 tag and a tobacco etch virus (TEV) cleavage site, and was expressed in Sf9 insect cells. Cells were collected by centrifugation (5000 × *g*, 10 min, 4 °C) and disrupted by sonication in buffer (50 mM Tris, pH 8.0, 150 mM NaCl) supplemented with 5.2 μg mL⁻¹ aprotinin, 2 μg mL⁻¹ leupeptin, and 1.4 μg mL⁻¹ pepstatin A (all from Calbiochem). Cell debris was removed by centrifugation (10,000 × *g*, 10 min, 4 °C). The membrane fraction was collected by ultracentrifugation (138,000 × *g*, 1 h, 4 °C). The membrane fraction was solubilized for 1 h at 4 °C in buffer (50 mM Tris, pH 8.0, 150 mM NaCl, 5 mM dithiothreitol, 1% digitonin (Calbiochem)). Insoluble materials were removed by ultracentrifugation (138,000 × *g*, 1 h, 4 °C). The detergent-soluble fraction was incubated with CNBr-Activated Sepharose 4 Fast Flow beads (GE Healthcare) coupled with an anti-GFP nanobody[28] and incubated for 1 h at 4 °C. The beads were washed with size-exclusion chromatography buffer (50 mM Tris, pH 8.0, 150 mM NaCl, 5 mM dithiothreitol, and 0.1% digitonin), and further incubated overnight with TEV protease to remove the GFP-His8 tag. After TEV protease digestion, the flow-through was collected, concentrated, and purified by size-exclusion chromatography on a Superose 6 Increase 10/300 GL column (GE Healthcare), equilibrated with size-exclusion chromatography buffer. The peak fractions of the protein were collected and concentrated to 3 mg mL⁻¹, using a centrifugal filter unit (Merck Millipore, 100 kDa molecular weight cutoff). A 3-μL portion of the concentrated HsLRRC8D protein was applied to a glow-discharged Quantifoil R1.2/1.3 Cu/Rh 300 mesh grid (Quantifoil), blotted using a Vitrobot Mark IV (FEI) under 4 °C and 100% humidity conditions, and then frozen in liquid ethane.

**EM image acquisition and data processing**. The grid images were obtained with a Tecnai Arctica transmission electron microscope (FEI) operated at 200 kV, and recorded by a K2 Summit direct electron detector (Gatan) operated in the super-resolution mode with a binned pixel size of 1.490 Å. The dataset was acquired with the SerialEM software[29]. Each image was dose-fractionated to 40 frames at a dose rate of 6–8 e⁻ pixel⁻¹ per second, to accumulate a total dose of ~50 e⁻ Å⁻². In total, 3397 super-resolution movies were collected. The movie frames were aligned in 5 × 5 patches, dose weighted and binned by two in MotionCor2[30], and down-sampled by two. Defocus parameters were estimated by CTFFIND 4.1[31]. First, template-based auto-picking was performed with the 2D class averages of a few hundred manually picked particles as templates[32]. A total of 2,056,807 particles were extracted in 3.25 Å pix⁻¹. These particles were divided into two batches and subjected to three rounds of 2D classification in RELION 2.1[32,33]. The initial model was generated in RELION. Subsequently, 1,062,374 good particles were further classified in 3D without symmetry. Finally, 247,154 particles in classes 6 and 7 with six intact subunits were re-extracted in the original pixel size of 1.49 Å pix⁻¹ and refined in C2 symmetry. The overall gold-standard resolution was 4.36 Å, with the local resolution in the core TM region extending to 3.9 Å and that in the peripheral region extending to 6.0 Å (Table 1 and Supplementary Fig. 2d).

**Model building**. De novo atomic modeling was conducted in COOT[34], using the HsLRRC8A (PDB ID: 5ZSU) and MmLRRC8A (PDB ID: 6G9O) structures as guides. The initial model was rebuilt using Rosetta[35] against the density maps, manually readjusted using COOT, and then refined using PHENIX[36] with secondary structure restraints.

**Mass spectrometry**. The purified HsLRRC8D protein (2.1 mg mL⁻¹, 50 μL) was reduced, carboxymethylated, desalted, and digested with trypsin (TPCK-treated, Worthington Biochemical Corporation). An aliquot of the digest was subjected to nano-liquid chromatography-tandem mass spectrometry, using a Q Exactive mass spectrometer (Thermo Fisher Scientific). The peptide mixtures (~1 μg) were separated on a nano ESI spray column (75 μm (inside diameter) × 100 mm (length), NTCC analytical column C18, 3 μm, Nikkyo Technos), with a gradient of 0–64% buffer B (acetonitrile with 0.1% (v/v) formic acid) in buffer A (MilliQ water with 0.1% (v/v) formic acid) at a flow rate of 300 nL min⁻¹ over 30 min (EASY-nLC 1000; Thermo Fisher Scientific). The mass spectrometer was operated in the positive-ion mode for mass spectrometry and tandem mass spectrometry, and the tandem mass spectrometry spectra were acquired using the data-dependent TOP 10 method. The tandem mass spectrometry data were used to search the NCBI-nr protein database (20160711, Taxonomy: H. sapiens 326,427 sequences) with the Mascot (version 2.6.0, Matrix Science) search engine, using the following parameters: enzyme = trypsin; maximum missed cleavages = 3; variable modifications = Acetyl (Protein N-term), Gln- > pyro-Glu (N-term Q), Oxidation (M); product mass tolerance = ±15 ppm; product mass tolerance = ±30 milli mass unit; instrument type = ESI-TRAP, and then quantified with the Proteome Discoverer 2.2 (Thermo Fisher Scientific) software package. All mass spectrometry data are presented in Supplementary Data 1 and have been deposited to the ProteomeXchange Consortium via the PRIDE partner repository with the dataset identifier PXD018780 and 10.6019/PXD018780.

**Cell culture for electrophysiology**. Human embryonic kidney HEK293 cells with the disruption of all five human *LRRC8* genes (*LRRC8⁻/⁻* cells)[4] were grown as a monolayer in Dulbecco's Modified Eagle's Medium (DMEM), supplemented with 10% fetal bovine serum, 40 units mL⁻¹ penicillin G, and 100 μg mL⁻¹ streptomycin under a 95% air, 5% CO₂ atmosphere at 37 °C. For electrophysiological experiments, cells were detached from the plastic substrate and cultured in suspension with agitation for 15–300 min.

**Patch-clamp experiments**. *LRRC8⁻/⁻* cells were grown to ~80% confluency (24–48 h after plating) in a 35-mm diameter dish. For patch-clamp experiments, cells were transfected using Lipofectamine 2000 (Thermo Fisher Scientific) transfection reagents with recombinant plasmids: pIRES2-AcGFP1-HsLRRC8A wild type, pIRES2-DsRed-HsLRRC8D wild type, pIRES2-DsRed-HsLRRC8D-F2C (F2C), pIRES2-DsRed-HsLRRC8D-T3C (T3C), pIRES2-DsRed-HsLRRC8D-L4C (L4C), pIRES2-DsRed-HsLRRC8D-A5C (A5C), pIRES2-DsRed-HsLRRC8D-E6C (E6C), pIRES2-DsRed-HsLRRC8D-V7C (V7C), pIRES2-DsRed-HsLRRC8D-A8C (A8C), pIRES2-DsRed-HsLRRC8D-S9C (S9C), and pIRES2-DsRed-HsLRRC8D-L10C (L10C). When co-transfecting plasmids for HsLRRC8A wild type and HsLRRC8D wild type, or for HsLRRC8A wild type and HsLRRC8D mutant, we selected a cell where both green and red fluorescent proteins were observed to confirm their simultaneous expression in the cell.

Whole-cell recordings were performed at room temperature (22–26 °C). Patch pipettes were pulled from borosilicate glass capillaries with a micropipette puller (P-97; Sutter Instruments, Novato, CA, USA). The electrode had a resistance of ~2–4 megaohms for whole-cell recordings when filled with pipette solution. Currents were recorded using an Axopatch 200B amplifier (Axon Instruments/Molecular Devices, Union City, CA, USA). Current signals were filtered at 5 kHz using a four-pole Bessel filter and digitized at 20 kHz. The pClamp software (version 10.5.1.0; Axon Instruments/Molecular Devices) was used for command pulse, data acquisition, and analysis. Data were analyzed using the Origin (OriginLab Corp., Northampton, MA, USA) software. For whole-cell recordings, the series resistance was compensated (to 70–80%) to minimize voltage errors. Ramp pulses were applied every 15 s from −100 mV to +100 mV at a speed of 0.8 mV ms⁻¹, from a holding potential ($V_h$) of 0 mV. The intracellular (pipette) solution contained (in mmol L⁻¹): 110 NMDG-Cl, 2 MgSO₄, 10 HEPES, 1 EGTA, and 2 Na₂ATP (pH adjusted to 7.3 with NMDG). The bath solution contained (in mmol L⁻¹): 110 NMDG-Cl, 5 MgSO₄, and 10 HEPES (pH adjusted to 7.4 with NMDG). For experiments on the effects of MTSES, the osmolality was adjusted with D-mannitol to 360 (hypertonic) and 330 (isotonic) mosmol kgH₂O⁻¹ in the pipette and bath solutions, respectively. For anion selectivity experiments, the osmolality was adjusted with D-mannitol to 300 mosmol kgH₂O⁻¹ in the pipette solution and 275 (hypotonic) and 330 (isotonic) mosmol kgH₂O⁻¹ in the bath solution. The time courses of current activation and inhibition were monitored by repetitively applying alternating step pulses to ±40 mV (500 ms duration, every 15 s) from a holding potential of 0 mV. For anion selectivity experiments, a 3-mol L⁻¹ KCl-agar salt bridge was used to minimize changes in the

liquid junction potentials when changing the bath solution from that containing 110 mmol L$^{-1}$ NMDG-Cl to that containing 110 mmol L$^{-1}$ NMDG-Br, NMDG-I, NMDG-glutamate, or NMDG-gluconate. The anion permeability ratio ($P_x/P_{Cl}$) was calculated from a shift in the reversal potential by the Goldman–Hodgkin–Katz Equation. The substituted-cysteine accessibility method (SCAM) analysis was performed with MTSES in LRRC8$^{-/-}$ cells expressing cysteine-substituted mutants of HsLRRC8D. MTSES (1 mmol L$^{-1}$) was added to the bath solution, after the whole-cell HsLRRC8A/D currents reached plateau values in the cells swollen by the hypertonic pipette solution.

**Statistics and reproducibility**. The sample sizes and statistical analyses used in patch-clamp recordings are presented in the legend of each figure.

**Reporting summary**. Further information on research design is available in the Nature Research Reporting Summary linked to this article.

## Data availability

The raw image of HsLRRC8D after motion correction has been deposited in the Electron Microscopy Public Image Archive (EMPIAR), under the accession number EMPIAR-10383. The cryo-EM density map of HsLRRC8D has been deposited in the Electron Microscopy Data Bank (EMDB), under the accession number EMD-30029. The molecular coordinates of HsLRRC8D have been deposited in the RCSB Protein Data Bank (PDB), under the accession number 6M04. All mass spectrometry data are available in Supplementary Data 1 and the ProteomeXchange Consortium with identifier PXD018780.

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

## Acknowledgements

We thank the members of the Nureki laboratory and Kikkawa laboratory, especially Dr. Keitaro Yamashita for technical assistance with the model building; Ms. Sanae Okazaki for technical assistance with the sample preparation; Mr. Atsuhiro Tomita for helpful comments on the manuscript; and Dr. Akihisa Tsutsumi and Dr. Haruaki Yanagisawa for optimization of the cryo-EM experiment. We also thank Dr. Takanori Nakane (MRC Laboratory of Molecular Biology) for technical assistance with EM image acquisition and data processing, and Dr. Koichi Nakajo (Jichi Medical University) for discussions and comments on the manuscript. This work was supported by a Ministry of Education, Culture, Sports, Science and Technology (MEXT) Grant-in-Aid for Specially Promoted Research (Grant No. 16H06294) to O.N.; and by JSPS KAKENHI (Grant Nos. 19K23833 to G.K., 18H03995 to H.I., and 19K16067 to K.W.); by the Project for Elucidating and Controlling Mechanisms of Aging and Longevity from Japan Agency for Medical Research and Development (AMED) (Grant No. JP17gm5010001) to H.I.; by the Platform Project for Supporting Drug Discovery and Life Science Research (Basis for Supporting Innovative Drug Discovery and Life Science Research (BINDS)) from AMED (Grant No. JP19am0101082) to M.S.; and by a grant from the RIKEN Dynamic Structural Biology project to M.S.

## Author contributions

G.K. and O.N. conceived and designed the project. R.N. and G.K. screened and purified the protein, and prepared cryo-EM samples. T.Numata and Y.O. designed the

patch-clamp experiments, T.Numata performed the experiments, and T.Numata, T.J.J., and Y.O. analyzed these data. N.D. assisted with the preparation of cryo-EM samples. T.Kato, T.H., M.I., K.W., and H.I. assisted with the functional characterization of samples. R.N., G.K., and T.Y. performed the cryo-EM data collection and processing, with assistance from M.S. T.Nishizawa, T.Kusakizako, and M.K. assisted with the optimization of the cryo-EM data collection. R.N., G.K., and R.I. performed the model building. R.N., T.Numata, G.K., T.J.J., Y.O., and O.N. wrote the manuscript. G.K., Y.O., and O.N. supervised all of the research.

## Competing interests

The authors declare no competing interests.
