## [Peer Review File · Communications Biology]

Reviewers' comments:

Reviewer #1 (Remarks to the Author):

Summary:

Herein, the authors report the structure and function of the human membrane protein LRRC8D - the largest one in the family of five proteins forming pores in VRAC channels. The VRAC-mediated transport of chloride ions and small organic compounds drives water efflux from the cell in response to cell swelling. In vivo, LRRC8A type protein forms hetero-oligomers (hexamers) with at least one more member of LRRC8 protein family which becomes a functional VRAC. The LRRC8 isoform combinations in VRACs determine channel properties such as substrate specificity and gating behavior.

When LRRC8A is combined with LRRC8D specifically, permeation of uncharged and negatively charged substrates is facilitated. The only existing structures available are those of the LRRC8A channel and although they provided substantial insight into the overall architecture of LRRC8 proteins, however these previous structures failed at capturing the structures of NTH and EL1 domains. These domains were found to be critical for many channel properties. The authors have thus investigated the LRRC8D protein with a focus on these domains.

A high-resolution, cryoEM analysis of homo-hexameric LRRC8D channel from Sf9 insect cells succeeded at capturing all parts of the protein. The structure in detergent (digitonin) has shown a dimer of trimers arrangement of all six subunits, not seen before with LRRC8 proteins. The channel is constricted at its extracellular side by EL1 domain residue F143, with the opening being wider than seen in the other published structures. Functional studies on WT and F143R mutant of LRRC8D suggested that the substitution from arginine (or lysine) residues in LRRC8A,B, C or E isoforms to phenylalanine residue in LRRC8D is important for increased permeability to a number of substrates observed in LRRC8D. No structural data for the F143R mutant was provided so it is not clear that this mutation changes the pore diameter itself. It is also not clear in what conformation the LRRC8D channel was captured in cryoEM map and what the pore diameter dynamics are during channel operation. Perhaps different conformational states could be captured in buffering conditions with different substrates present.

The NTH pore-lining domain found in cryoEM was confirmed to exist in hetero-hexameric LRRC8A+D channels with electrophysiology experiments on NTH cysteine mutants. More research needs to be done to verify the role of each pore-facing NTH residues in different channel properties. The authors could have expanded the functional analysis of select, pore-lining, NTH cysteine mutants to the included substrate permeability as was done for F143R mutant which would have made for a more complete investigation. In addition, the stoichiometry of hetero-oligomer formation in LRRC8 proteins should be examined and in the future as well as their structures when possible.

The obvious concern with the structural models determined so far (homo-hexamers) is that the homo-hexamers have no transport activity and it is the hetero-hexamers that function to transport the organic molecules (at least in the functional assays performed). Thus, insight into why hetero-hexamers are active and homo-hexamers are not will be an important future contribution to understanding transport.

However this study does provide important insight into two newly resolved regions in the LRRC8 proteins, I recommend this work for publication after addressing the issues presented below.

Suggestions:

Line 95: For completeness, it would be best to include the control condition with all 5 LRRC8 genes knocked out if possible as it is mentioned in the text that HsLRRC8A currents were 'slight' even though the Sup. Fig 1 d) panel HsLRRC8A looks almost entirely flat – is the slight current observed relative to the knockout control? Maybe a close-up view of your current trace could be useful to visualize small changes?

Line 174: You say that the F143R mutation in LRRC8D 'tightens the pore size at the constricted site on the extracellular side'. Evidence for that in your electrophysiology experiments should be cited here in the absence for structural data for that mutant.

The F143R mutant seems to reduce permeability of negatively charged substrates – such reduction in anion permeability could be expected from mutations to negatively charged amino acid side chains, but it is harder to explain with a mutation to arginine. Please explain.

It may also be clearer if you describe the experiments of the other research group on positive charge of R103. Mention that R103 in LRRC8A is at equivalent position to F143 in LRRC8D (Point to your Fig. 3 c)).

Line 235/238: Correct the formatting of the reference given.

Sup. Fig. 2 g) and Sup. Table 1: Why are you comparing a final protein model to its various maps and then report on the model resolution by assuming FSC cutoff of 0.5? Typically, a model is built to a map (with a reported resolution, usually heterogenous) and the analysis of model-to-map fit is performed by showing figures of regions with different resolutions.

Fig. 2: This figure would benefit from a comparison(s) between a structure of a monomer of LRRC8A with a monomer of LRRC8D- it could be prepared similar to how Fig. 1 b) looks like but using a graphical representation/style of Fig. 2, with two monomer structures overlaid.

Fig. 3 or Sup. Fig 4: Consider adding analyses of channel pore diameter with a software like HOLE.

Reviewer #2 (Remarks to the Author):

This article by Nakamura et al., reported a cryoEM structure of LRRC8D homohexmer. The structure revealed a wider opening on the extracellular side as well as an N-terminal helix that protrudes into the pore from the intracellular side. The authors then used electrophysiology to assess the potential relevance of the wider pore and to validate the location of the N-terminal helix. This article is well written and could be a valuable addition to the field. I would recommend its publication if the following points are addressed or discussed.

1. In Fig 3D, IV traces of F143R have greater slopes for all anions than the traces of WT, which suggests that F143R has a higher open probability or a larger conductance. Although not always true, but if Phe143 creates a wider opening for larger anions to permeate, one would imagine that the conductance of the WT should be larger than F143R. What are the authors' opinions on this? If the F143R also affects channel open probability, could the author provide any hypothesis?
2. In Fig 4C, only A8C has an asterisk symbol. Is this correct? S9C and E6C (maybe also F2C) seem to

be significantly different from WT.

3. Glu6 appears to face the permeation pathway in the structure and the authors also discussed that Glu6 is important for anion selectivity in lines 240. However, this position can not be modified by MTSES (Fig 4E). Could the authors provide an explanation? Or at least describe this in the text to disclose the fact.

4. The authors do not have to do this experiment but I am wondering if the C6 symmetry is applied to the TM region, will the density of N-terminal helix further improve.

Reviewer #1 comments: Nakamura *et al.* COMMSBIO-19-1887-T

“Reviewers' comments:

Reviewer #1 (Remarks to the Author):

Summary:

Herein, the authors report the structure and function of the human membrane protein LRRC8D - the largest one in the family of five proteins forming pores in VRAC channels. The VRAC-mediated transport of chloride ions and small organic compounds drives water efflux from the cell in response to cell swelling. In vivo, LRRC8A type protein forms hetero-oligomers (hexamers) with at least one more member of LRRC8 protein family which becomes a functional VRAC. The LRRC8 isoform combinations in VRACs determine channel properties such as substrate specificity and gating behavior.

When LRRC8A is combined with LRRC8D specifically, permeation of uncharged and negatively charged substrates is facilitated. The only existing structures available are those of the LRRC8A channel and although they provided substantial insight into the overall architecture of LRRC8 proteins, however these previous structures failed at capturing the structures of NTH and EL1 domains. These domains were found to be critical for many channel properties. The authors have thus investigated the LRRC8D protein with a focus on these domains.

A high-resolution, cryoEM analysis of homo-hexameric LRRC8D channel from Sf9 insect cells succeeded at capturing all parts of the protein. The structure in detergent (digitonin) has shown a dimer of trimers arrangement of all six subunits, not seen before with LRRC8 proteins. The channel is constricted at its extracellular side by EL1 domain residue F143, with the opening being wider than seen in the other published structures. Functional studies on WT and F143R mutant of LRRC8D suggested that the substitution from arginine (or lysine) residues in LRRC8A,B, C or E isoforms to phenylalanine residue in LRRC8D is important for increased permeability to a number of substrates observed in LRRC8D. No structural data for the F143R mutant was provided so it is not clear that this mutation changes the pore diameter itself. It is also not clear in what conformation the LRRC8D channel was captured in cryoEM map and what the pore diameter dynamics are during channel operation. Perhaps different conformational states could be captured in buffering conditions with different

substrates present.

The NTH pore-lining domain found in cryoEM was confirmed to exist in hetero-hexameric LRRC8A+D channels with electrophysiology experiments on NTH cysteine mutants. More research needs to be done to verify the role of each pore-facing NTH residues in different channel properties. The authors could have expanded the functional analysis of select, pore-lining, NTH cysteine mutants to the included substrate permeability as was done for F143R mutant which would have made for a more complete investigation. In addition, the stoichiometry of hetero-oligomer formation in LRRC8 proteins should be examined and in the future as well as their structures when possible.

The obvious concern with the structural models determined so far (homohexamers) is that the homohexamers have no transport activity and it is the heterohexamers that function to transport the organic molecules (at least in the functional assays performed). Thus, insight into why heterohexamers are active and homohexamers are not will be an important future contribution to understanding transport.

However this study does provide important insight into two newly resolved regions in the LRRC8 proteins, I recommend this work for publication after addressing the issues presented below.”

We appreciate your positive comments regarding our manuscript. The specific points mentioned by Reviewer #1 are addressed below.

“Suggestions:

Line 95: For completeness, it would be best to include the control condition with all 5 LRRC8 genes knocked out if possible as it is mentioned in the text that HsLRRC8A currents were ‘slight’ even though the Sup. Fig 1 d) panel HsLRRC8A looks almost entirely flat – is the slight current observed relative to the knockout control? Maybe a close-up view of your current trace could be useful to visualize small changes?”

Thank you for this comment. Accordingly, we have now included the control data recorded in *LRRC8*^{-/-} cells as an Inset in **Supplementary Fig. 1 (d)** to highlight the close-up LRRC8A currents. As indicated in **Supplementary Fig. 1e**, the LRRC8A-expressing cells exhibited significant currents, in contrast to the control cells, although they were much smaller than the LRRC8A/D currents.

“Line 174: You say that the F143R mutation in LRRC8D ‘tightens the pore size at the constricted site on the extracellular side’. Evidence for that in your electrophysiology experiments should be cited here in the absence for structural data for that mutant.”

Thank you for this comment. Accordingly, we have stated “thus exhibiting a greater effect on tightening the pore size than on changing the electrostatic effect, thereby leading to a reduction, rather than an increase, in the permeability to glutamate and gluconate (Fig. 3f)” in the revised manuscript (Page 8, Lines 176-179).

“The F143R mutant seems to reduce permeability of negatively charged substrates – such reduction in anion permeability could be expected from mutations to negatively charged amino acid side chains, but it is harder to explain with a mutation to arginine. Please explain.”

Thank you for this comment. Our results suggested that the F143R mutation changes both the electrostatic effects and pore size at the constricted site on the extracellular side. Although the electric charge and bulkiness are both important factors, the F143R mutation might have exhibited a greater effect on tightening the pore size than on changing the electrostatic effect, thereby reducing the permeability to the tested substrates in descending order. We have thus added descriptions about these points, as given above, in the revised manuscript (Page 8, Lines 176-179), and followed by “However, further structural analyses are required to understand the precise effect of the mutation from phenylalanine to arginine, since the structure of the HsLRRC8D F143R mutant has not been determined” (Pages 8-9, Lines 179-181).

“It may also be clearer if you describe the experiments of the other research group on positive charge of R103. Mention that R103 in LRRC8A is at equivalent position to F143 in LRRC8D (Point to your Fig. 3 c).”

Thank you for this comment. Accordingly, we carefully examined the electrophysiological analyses of the LRRC8A R103 mutant performed by Deneka *et al.*, *Nature* 2018 and Kefauver *et al.*, *eLife* 2018, and have stated that “Notably, previous reports indicated

that the positive charge and the bulkiness of Arg103 in LRRC8A may be involved in the anion selectivity (Deneka *et al.* 2018) and the pore blockage by extracellular ATP (Kefauver *et al.* 2018), respectively.” in the revised manuscript (Page 9, Lines 181-183).

“Line 235/238: Correct the formatting of the reference given.”

According to the comment, we have corrected the reference format (Page 12, Lines 243, 246, and 249).

“Sup. Fig. 2 g) and Sup. Table 1: Why are you comparing a final protein model to its various maps and then report on the model resolution by assuming FSC cutoff of 0.5? Typically, a model is built to a map (with a reported resolution, usually heterogenous) and the analysis of model-to-map fit is performed by showing figures of regions with different resolutions.”

We appreciate this comment and apologize for our poor explanation. In **Supplementary Fig. 2g**, we showed FSC plots of refined atomic models against the cryo-EM density maps to detect possible overfitting of the models to the density maps, but we agree that showing panels of different regions would be more useful since our HsLRRC8D structure has a wide resolution range. In the revised figure, we have deleted **Supplementary Fig. 2g** and added some panels showing regions of the structure models with the density maps in **Supplementary Fig. 2e**.

“Fig. 2: This figure would benefit from a comparison(s) between a structure of a monomer of LRRC8A with a monomer of LRRC8D- it could be prepared similar to how Fig. 1 b) looks like but using a graphical representation/style of Fig. 2, with two monomer structures overlaid.”

Thank you for this comment. Accordingly, we have prepared the new **Fig. 2c** presenting the superimposition of the HsLRRC8D and HsLRRC8A monomers with cylinder representations.

“Fig. 3 or Sup. Fig 4: Consider adding analyses of channel pore diameter with a software

like HOLE.”

Thank you for this comment. Accordingly, we have calculated the pore diameters of the HsLRRC8D and HsLRRC8A structures using the HOLE program, and presented them as the new **Fig. 3c**.

Reviewer #2 comments: Nakamura *et al.* COMMSBIO-19-1887-T

“Reviewer #2 (Remarks to the Author):

This article by Nakamura et al., reported a cryoEM structure of LRRC8D homohexmer. The structure revealed a wider opening on the extracellular side as well as an N-terminal helix that protrudes into the pore from the intracellular side. The authors then used electrophysiology to assess the potential relevance of the wider pore and to validate the location of the N-terminal helix. This article is well written and could be a valuable addition to the field. I would recommend its publication if the following points are addressed or discussed.”

We appreciate your positive comments regarding our manuscript. The specific points mentioned by Reviewer #2 are addressed below.

“1. In Fig 3D, IV traces of F143R have greater slopes for all anions than the traces of WT, which suggests that F143R has a higher open probability or a larger conductance. Although not always true, but if Phe143 creates a wider opening for larger anions to permeate, one would imagine that the conductance of the WT should be larger than F143R. What are the authors’ opinions on this? If the F143R also affects channel open probability, could the author provide any hypothesis?”

Thank you for this comment. The data presented in **Fig. 3d** (**Fig. 3e** in the revised manuscript) represent the whole-cell currents recorded only in one of six LRRC8A/D-expressing cells and one of six LRRC8A/D-F143R-expressing cells. The mean values of conductance evaluated at their reversal potentials were 72.5 ± 15.8 nS and 50.4 ± 6.9 nS for 8A/D and 8A/D-F143R cells, respectively, and are not statistically different from each other. The whole-cell current ($I = n \times P_o \times i$) is determined not only by the open probability (P_o) and the single-channel current (i), which is proportional to the single-channel conductance, but also by the number of channels expressed in given cells (n), which inevitably varies from cell to cell. We preliminarily examined the expression levels of LRRC8A/D and LRRC8A/D-F143R using fluorescence-detection size-exclusion chromatography (FSEC) (Kawate *et al.*, *Structure* 2006), and found that the expression level of LRRC8A/D-F143R is higher than that of LRRC8A/D (as shown below in panels **a** and **b**, and error bars in **b** indicate \pm s.e.m. for $n = 3$). Therefore, when

the P_o values of both channels can be assumed to be identical, the single-channel conductance would be higher in LRRC8A/D than in LRRC8A/D-F143R, which is consistent with our hypothesis that Phe143 creates a wider opening for larger anion

permeation. Nevertheless, precise evaluations of the single-channel conductance and the open probability await future single-channel recording studies.

“2. In Fig 4C, only A8C has an asterisk symbol. Is this correct? S9C and E6C (maybe also F2C) seem to be significantly different from WT.”

Thank you for this comment. Accordingly, for **Fig. 4c**, we have now increased the number of experiments from $n=5$ to $n=8-9$. However, our ANOVA analysis has again shown statistical significance only for A8C, as indicated in the revised **Fig. 4b**.

“3. Glu6 appears to face the permeation pathway in the structure and the authors also discussed that Glu6 is important for anion selectivity in lines 240. However, this position can not be modified by MTSES (Fig 4E). Could the authors provide an explanation? Or at least describe this in the text to disclose the fact.”

Thank you for this comment. The previous electrophysiological analyses by Zhou *et al.*, *J. Biol. Chem.* 2018 demonstrated that double cysteine mutations at Glu6 in LRRC8A/C (“E,E6C” shown in panel D, modified from the original Zhou *et al.* Figure 3D) increased P_i/P_{Cl} as compared to WT, while the same mutants in LRRC8A/D and

LRRC8A/E also increased P_1/P_{Cl} but to lesser extents (panel G modified from original Zhou *et al.* Figure 3G) (see above panel). These results suggest that the sequence diversity in the N terminal regions influences the substrate accessibility, which affects the reactivity to MTSES among different isoform combinations. We have added descriptions about these points in the revised manuscript (Page 12, Lines 249-253).

“4. The authors do not have to do this experiment but I am wondering if the C6 symmetry is applied to the TM region, will the density of N-terminal helix further improve.”

Thank you for this comment. Accordingly, we performed a focused refinement of the TM region with C6 symmetry imposed, but the density of the N-terminal helix was not improved due to the overfitting (right panel).

REVIEWERS' COMMENTS:

Reviewer #1 (Remarks to the Author):

We have no further comments, the authors have addressed all of our concerns.

Reviewer #2 (Remarks to the Author):

The authors have addressed all my questions. I would recommend the publication of the manuscript